# Overweight and Obesity among Adults in Iraq: Prevalence and Correlates from a National Survey in 2015

**DOI:** 10.3390/ijerph18084198

**Published:** 2021-04-15

**Authors:** Supa Pengpid, Karl Peltzer

**Affiliations:** 1ASEAN Institute for Health Development, Mahidol University, Salaya, Phutthamonthon, Nakhon Pathom 73170, Thailand; supa.pen@mahidol.ac.th; 2Department of Research Administration and Development, University of Limpopo, Sovenga 0727, South Africa; 3Department of Psychology, University of the Free State, Bloemfontein 9300, South Africa

**Keywords:** overweight, obesity, health behaviour, health status, adulthood

## Abstract

This study aimed to estimate the prevalence and correlates of overweight and obesity among adults in Iraq. Data from a 2015 nationally representative cross-sectional survey of 3916 persons 18 years or older (M (median) age = 40 years, IQR (interquartile range) age = 29–52 years; men: M = 41 years, IQR = 29–54 years; women: M = 40 years, IQR = 30–51 years) who responded to a questionnaire, and physical and biochemical measures were analysed. Multinomial logistic regression was utilised to predict the determinants of overweight and obesity relative to under or normal weight. The results indicate that 3.6% of the participants were underweight (body mass index (BMI) <18.5 kg/m^2^), 30.8% had normal weight (BMI 18.5–24.9 kg/m^2^), 31.8% were overweight (25.0–29.9 kg/m^2^), and 33.9% had obesity (BMI ≥30.0 kg/m^2^). In the adjusted multinomial logistic regression, being aged 40–49 years (compared to 18–39 years old) (adjusted relative risk ratio (ARRR): 4.47, confidence interval (CI): 3.39–5.91), living in an urban residence (ARRR: 1.28, CI: 1.14–2.18), and having hypertension (ARRR: 3.13, CI: 2.36–4.17) were positively associated with obesity. Being male (ARRR: 0.47, CI: 0.33–0.68), having more than primary education (ARRR: 0.69, CI: 0.50–0.94), and having a larger household size (five members or more) (ARRR: 0.45, CI: 0.33–0.60) were negatively associated with obesity. Approximately two in three adult participants were overweight/obese, and sociodemographic and health risk factors were found that can be utilised in targeting interventions.

## 1. Introduction

More than half (55%) of all mortality in Iraq in 2016 was attributable to non-communicable diseases (NCDs) [1]. Most NCDs result from poor diet, physical inactivity, tobacco use, and harmful alcohol use leading to metabolic/physical changes, including hypertension, diabetes, and overweight and obesity [2]. Worldwide, among adults, the prevalence of obesity (body mass index (BMI) ≥30 kg/m^2^) is 10.8% among men and 14.9% among women [3]. In several local surveys in subregions and clinical populations in Iraq, high proportions of obesity have been reported. For example, in a community-based survey (*N* = 1480 adults in 2017) in Erbil city, Iraq, the prevalence of overweight was 33.4% and obesity 40.9% [4], and in Basrah, Southern Iraq (2003–2010), overweight/obesity was 55.1% [5]. Among nonpregnant women (*N* = 200, ≥18 years) attending outpatient clinics in Baghdad, Iraq, 39% were overweight and 37% had obesity [6], and among female relatives of primary care attendees (*N* = 440) in Baghdad, the prevalence of obesity was 35.2% [7]. In a national STEPwise approach to surveillance (STEPS) survey in 2005–2006 in Iraq (25–65 years), the prevalence of overweight/obesity was 66.9% [8]. To the best of our knowledge, there are no recent national adult data on the prevalence and correlates of overweight and obesity in Iraq. To better plan interventions, national estimates and risk factors of overweight and obesity are needed in Iraq.

In the Eastern Mediterranean region, the prevalence of overweight/obesity among adults ranges from 25% to 81.9% [9]. In 2014, in Kuwait, the adult (18–69 years) prevalence of overweight was 37% and obesity 40.3% [10], while in 2016, in Iran, the prevalence of overweight/obesity (BMI ≥25 kg/m^2^) was 59.3% [11]. In 2017, in Jordan, overweight or obesity (BMI ≥25 kg/m^2^) was 77.2% among men and 74.5% among women (≥18 years) [12], and in Morocco, overweight was 35.5% and obesity 20.6% (≥18 years) [13].

Possible risk factors for obesity in the Eastern Mediterranean region may include dietary changes, sedentary lifestyle, stunting, promotion of high-fat foods, and perceived body image [9]. Moreover, the odds for overweight/obesity may increase in middle age [14,15] and older age [4,5,6,7], among women [4,5,15,16], those with higher socioeconomic status [14,15,17], illiterate women [5], those ever married [4], and those with urban residence [15,16,17,18]. Some studies have shown that tobacco use is inversely associated with overweight/obesity [16,19,20], while poor dietary behaviour, such as intake of foods high in fat and sugars or insufficient fruit and vegetables and physical inactivity are positively associated with overweight/obesity [19,20,21,22,23]. Other studies have shown an association between overweight/obesity and NCDs, such as hypertension and diabetes [24,25]. In a systematic review, depression increased the odds for developing obesity [26]. The study aimed to estimate the national prevalence and correlates of overweight and obesity among adults in Iraq.

## 2. Methods

This is a secondary analysis conducted using nationally representative population-based and cross-sectional data from the 2015 Iraq STEPS survey [27]. The data and more detailed sampling methods can be accessed [27]. The study response rate was more than 93% [27,28]. Briefly, a multistage cluster sampling was used with stratification to urban and rural areas. Primary sampling units (PSUs) (*N* = 412) were the blocks, which consisted of 70 households or more before selection. One person from each household was randomly selected. In total, 4071 persons 18 years or older were potentially eligible in this study. However, 155 individuals were excluded (124 were pregnant and 31 did not have complete anthropometric measurements), so that 3916 participants were included in the final data analysis.

Sample size calculation. Based on the 2005–2006 Iraq national STEPS survey, the prevalence of overweight/obesity was 66.9% [8]. Based on this information, the sample size was calculated with an expected overweight/obesity prevalence of 66%, an acceptable margin of 5%, and clusters of 412. The minimum sample for each cluster was 2, and the minimum sample was 824. In this study, we used all 3916 participants for the analysis.

Ethical approval for the study was obtained from the Republic of Iraq Ministry of Health/Environment Public Health Directorate, and written informed consent was obtained from the participants [28].

### 2.1. Measures

The standardised anthropometric measuring devices (Uniscale weighing scale and seca measuring tapes for height and waist circumference) available at the Nutrition Research Institute in Baghdad and the related nutrition units in the governorates were utilised. The trained teams were provided with checklists for correct physical measurements [28]. BMI was classified as “<18.5 kg/m^2^ underweight, 18.5–24.4 kg/m^2^ normal weight, 25–29.9 kg/m^2^ overweight and ≥30 kg/m^2^ obesity” [29]. Central or abdominal obesity was defined as “waist circumference ≥102 cm for males and ≥88 cm for females” [30].

Hypertension or raised blood pressure (BP) was defined as “systolic BP ≥140 mm Hg and/or diastolic BP ≥90 mm Hg, or where the participant is currently on antihypertensive medication” [31], based on the average of the last two of three BP readings. After resting for 15 min, the first BP reading was taken, and the second and third BP readings were taken in 3 min spacings. The auscultatory method of BP measurement with a properly calibrated and validated mercury type of sphygmomanometer utilised by a medical assistant or nurse [28].

Diabetes was defined as “fasting plasma glucose levels ≥7.0 mmol/L (126 mg/dL), or using insulin or oral hypoglycaemic drugs” [32]. Blood samples were drawn in the morning (after 10–14 h fasting). Diabetic patients on medication were asked to bring their tablets and insulin with them and to take them after their blood measurement

The level of fasting plasma glucose was determined using the enzymatic method (glucose oxidase) [28].

History of cardiovascular disorder was asked with questions on having had a heart attack and/or a stroke (Yes, No) [28].

The health risk behaviour variables comprised: ever having used alcohol, “Have you ever consumed any alcohol such as beer, wine, spirits, etc.?” (Yes, No); exposure to secondary smoke in the past month (closed spaces at work or at home) and smoking status, “Do you currently smoke any tobacco products, such as cigarettes, cigars or pipes?” (Yes, No) and “In the past, did you ever smoke any tobacco products?” (Yes, No) [28]. Dietary behaviour included daily fruit and vegetable consumption measured from the total number of servings of fruit and vegetables eaten per day in a typical week [28]. Inadequate fruit and vegetable intake was classified as having less than 5 servings a day [33]. Meals outside the home were assessed with the question, “On average, how many meals per week do you eat that were not prepared at home? By meal, I mean breakfast, lunch and dinner?” (Number) [28]. Sedentary behaviour (≥8 h/day) [34]; low, moderate, or high physical activity were assessed with the Global Physical Activity Questionnaire [35]. “High physical activity: A person reaching any of the following criteria is classified in this category: Vigorous-intensity activity for at least 3 days achieving a minimum of at least 1500 MET (metabolic equivalent) minutes per week; 7 or more days of any combination of walking, moderate or vigorous intensity activities achieving a minimum of at least 3000 MET-minutes per week. Moderate physical activity: A person not meeting the criteria for the high category, but meeting any of the following criteria is classified in this category: 3 or more days of vigorous-intensity activity of at least 20 min per day; 5 or more days of moderate-intensity activity or walking of at least 30 min per day; 5 or more days of any combination of walking, moderate or vigorous intensity activities achieving a minimum of at least 600 MET-minutes per week. Low physical activity: Person not meeting any of the above mentioned criteria falls in this category.” [35].

Sociodemographic information included age, sex, highest educational level, number of adult household members, and residential status [28]. Household crowding has been found to have an inverse relationship with socioeconomic status [36].

### 2.2. Data Analysis

All statistical procedures were adjusted for complex sample design and conducted with Stata software version 13.0 (Stata Corporation, College Station, TX, USA). The data were weighted “to make the sample representative of the target population in Iraq (by sex and by age group: 18–39, 40–59, 60 and over).” [28]. The chi-square test calculated the differences in proportions. Multinomial logistic regression was used to estimate the predictors of overweight and obesity (with BMI <25 kg/m^2^ as the reference category). The underweight and normal-weight subjects were combined because of the small underweight group. No multi-collinearity was detected. We also tested for possible interactions among other variables such as age group and sex, and overweight and obesity. However, we did not find any significant interaction effects and omitted them from the final model. Missing values (<5%) were excluded from the analysis. *P* < 0.05 was considered significant.

## 3. Results

### Sample and Body Mass Index Information

The sample consisted of 3916 individuals aged 18 years and older (M (median) age = 40 years, IQR (interquartile range) age = 29–52 years; men: M = 41 years, IQR = 29–54 years; women: M = 40 years, IQR = 30–51 years). Nearly 60% (59.2%) were female. More than half of the participants (51.5%) were living with five or more adult household members, 37.2% had less than primary education, and 23.7% lived in rural areas. Four in five participants (79.1%) consumed inadequate fruit and vegetables, 47.3% had eaten at least one meal outside home in the past week, 52.2% engaged in low physical activity, and 27% had high sedentary behaviour (≥8 h/day). More than one in five respondents (21.3%) were currently smoking, 60.6% were exposed to secondary smoke, and 2.4% had drunk alcohol before. The proportion of participants with hypertension was 36.2%, while 14% had type 2 diabetes, and 4.5% reported a history of heart attack or stroke. In all, 3.6% of the study sample was underweight (BMI <18.5 kg/m^2^), 30.8% had normal weight (BMI 18.5–24.9 kg/m^2^), 31.8% were overweight (25.0–29.9 kg/m^2^), and the obesity rate was 33.9%. Further sample details are listed in Table 1 below.

The highest proportion of underweight (6.9%) was in the 18 to 29 age group (8.5% of men and 4.6% of women), while the highest proportion of general and central obesity was in the 45 to 59 age group (64.2% general obesity among women and 41.4% among men; 91.8% central obesity among women and 56.4% among men). See Table 2.

## 4. Multinomial Logistic Regression for Overweight and Obesity

Factors positively associated with obesity were being aged 40–49 years (compared to 18–39 years old) (adjusted relative risk ratio (ARRR): 4.47, confidence interval (CI): 3.39–5.91, *p* < 0.001), living in an urban residence (ARRR: 1.28, CI: 1.14–2.18, *p* = 0.006), and having hypertension (ARRR: 3.13, CI: 2.36–4.17, *p* < 0.001). Factors negatively associated with obesity were being male (ARRR: 0.47, CI: 0.33–0.68, *p* < 0.001), having more than primary education (ARRR: 0.69, CI: 0.50–0.94, *p* = 0.020), and having a larger household size (five members or more) (ARRR: 0.45, CI: 0.33–0.60, *p* < 0.001). Apart from educational level, all of these associations were also found for overweight. See Table 3.

## 5. Discussion

In this national 2015 Iraq STEPS survey, the found prevalence of overweight (31.8%, BMI ≥25.0–29.9 kg/m^2^) and obesity (33.9%, BMI ≥30.0 kg/m^2^) or overweight/obesity (65.7%) seems very similar to previous local investigations, e.g., in Erbil city [4], in Basrah [5], among females in out-patient clinics in Baghdad [6], among female relatives of primary care attendees in Baghdad [7], in the 2005–2006 Iraq national STEPS survey (66.9% overweight or obesity) [8], and in Kuwait (37% overweight and 40.3% obesity) [10]. However, it was lower than in Jordan (>75% overweight or obesity) [12] and higher than in Iran (59.3%) [11], Morocco (35.5% overweight and 20.6% obesity) [13], and global estimates (10.8% of men and 14.9% of women obesity) [1]. The high prevalence of obesity in this study may have contributed to the high prevalence of diabetes (14%) and hypertension (36.2%). Generally, the high prevalence of overweight and obesity in Iraq might be attributed to a continued demographic and epidemiological transition, economic improvement, and redistribution of wealth after political changes [7].

Consistent with previous studies [14,15,16,17], this study found that being female, being middle-aged, having a higher socioeconomic status (less household crowding), and residing in urban areas were associated with being overweight and/or having obesity. Obesity interventions may be reinforced by targeting women, the middle-aged, those with a higher socioeconomic status, and people residing in urban areas. Of concern as well is that 55.2% of young women and 40% of young men aged 18–29 were already overweight or obese, showing that a large proportion of overweight/obesity is already established in early adulthood. Therefore, obesity interventions starting in childhood or adolescence should be prioritised in Iraq [10]. One additional factor contributing to a higher rate of obesity among women than men may be related to cultural restrictions limiting access to exercise [12]. Compared to people aged 18–39, those aged 40–49 had significantly higher odds of obesity, but this was no longer significant for persons 50 years and older in this study. Individuals 50 years and older in this study may have been 40–49 during the time of the United Nations sanctions (UNS) (1991–2002), which was accompanied by a dramatic decrease in free-sugar consumption and a decrease in overweight and obesity [37] that some may have maintained as they grew older. This study found that higher education was protective against obesity, which is in line with a study in Kuwait [10] and a review showing that in middle-income countries, “education may protect against the obesogenic effects of increased household wealth as countries develop” [38]. People with higher education may be more concerned with their health and may, consequently, adopt a healthier lifestyle [12].

This study did not find an association between dietary behaviour (inadequate fruit and vegetable intake and having meals outside home) and overweight or obesity, unlike some previous research [19,20,21]. This study did not assess other dietary behaviours, such as frequent snacking, skipping breakfast, eating high amounts of processed or fast food, and high intake of sugary beverages, which may have been responsible for a higher rate of overweight/obesity [9]. In agreement with previous studies [4,10,14,16,19,20,22,23], this study showed via bivariate analysis that current smoking, passive smoking, and physical activity were inversely associated with overweight and obesity. Smoking may act on body weight “by increasing energy expenditure and inhibiting the expected compensatory increase in caloric intake” [39]. As shown previously [10,24,25], we found an association between NCDs (hypertension and, in univariate analysis diabetes) and overweight/obesity. Some of the recommendations by the STEP report for Iraq include dietary and physical activity intervention, including “Promotion of urban planning and transportation policies supportive of physical activity; implementation of setting-based physical activity programmes at school and workplaces; enactment of nutritional policies for food products marketing, and initiation of indoor programmes on healthy diet and physical exercise targeting the female and the elderly” [28].

## 6. Study Limitations

The strength of this study was the nationally representative sample using standardised measures. Apart from physical and biomedical measures, self-reported questionnaire data may have suffered from biased responses. Another limitation was the cross-sectional nature of the survey, which does not allow for causative conclusions. Inter-observer variation was not calculated for the physical measurements, which is a further study limitation. Some variables affecting body weight status, e.g., stress and mental health [26], assessed in the 2015 Iraq STEPS survey were not available in the publicly available dataset, and therefore could not be included.

## 7. Conclusions

This study found that in the 2015 adult national Iraq STEPS survey, approximately two in three participants were overweight/obese. Several risk factors, including sociodemographics (middle age (40–49 years old), female sex, lower education, higher socioeconomic status, and urban residence) and hypertension, were identified for overweight and/or obesity, which can be targeted in interventions. Implementing preventive interventions, such as programmes improving a healthy diet, appropriate food policies, promotion of physical activity and interrupting sedentary behaviour, and community awareness campaigns may help in ameliorating the high burden of overweight and obesity. The evaluation of experimental weight reduction interventions is recommended as future research to fine-tune intervention strategies in Iraq.

## Figures and Tables

**Table 1 ijerph-18-04198-t001:** Sample and bodyweight classification by sociodemographic and health variables among adults in Iraq, 2015.

Variable (#Missing Values)	Sample	Underweight	Normal Weight	Overweight	Obesity	*p*-Value
	*N* (%)	*N* (%)	*N* (%)	*N* (%)	*N* (%)	
All	3916	87 (3.6)	912 (30.8)	1279 (31.8)	1638 (33.9)	
Age in years (#10)						<0.001
18−29	956 (44.4)	55 (6.9)	395 (46.8)	286 (27.6)	220 (18.7)	
30−44	1389 (26.6)	17 (1.1)	255 (19.8)	485 (37.2)	632 (41.8)	
45−59	927 (18.5)	7 (0.7)	122 (12.8)	299 (33.3)	499 (53.2)	
60−69	634 (10.5)	8 (1.1)	138 (22.3)	208 (33.2)	280 (43.3)	
Sex (#0)						<0.001
Female	2318 (59.2)	35 (2.2)	445 (24.5)	705 (30.3)	1133 (43.0)	
Male	1598 (40.8)	52 (4.8)	467 (36.2)	574 (33.1)	505 (25.9)	
Education (#22)						<0.001
<Primary	1683 (37.2)	28 (2.4)	367 (26.8)	500 (29.9)	788 (41.0)	
Primary	989 (24.4)	20 (2.9)	219 (30.1)	349 (33.6)	401 (33.4)	
>Primary	1222 (38.4)	39 (5.3)	321 (35.2)	421 (32.2)	441 (27.2)	
Adult household members (#5)						<0.001
1−2	1390 (17.3)	20 (1.5)	268 (20.7)	460 (34.4)	642 (43.4)	
3−4	1355 (31.2)	28 (3.3)	321 (28.8)	445 (32.4)	561 (35.6)	
≥5	1166 (51.5)	39 (4.6)	323 (35.4)	372 (30.5)	432 (29.6)	
Residence (#0)						0.022
Rural	832 (23.7)	19 (3.7)	248 (37.0)	244 (28.9)	321 (30.4)	
Urban	3084 (76.3)	68 (3.6)	664 (28.8)	1035 (32.7)	1317 (34.9)	
Fruit and vegetable consumption (#18)						0.636
≥5 servings	870 (20.9)	22 (3.5)	187 (28.9)	295 (32.3)	366 (35.3)	
<5 servings	3028 (79.1)	65 (3.7)	720 (31.2)	979 (31.6)	1264 (33.5)	
Meals outside home (#92)						0.005
0	2232 (52.7)	51 (3.7)	501 (29.5)	717 (31.0)	963 (35.8)	
1	778 (20.3)	16 (4.0)	172 (27.2)	268 (34.7)	322 (34.1)	
≥2	814 (27.0)	20 (3.5)	228 (37.0)	257 (30.7)	309 (28.8)	
Physical activity (#3)						<0.001
Low	2174 (52.2)	50 (3.3)	463 (28.0)	671 (30.2)	990 (38.5)	
Moderate	906 (22.6)	15 (3.2)	214 (28.0)	304 (33.7)	373 (35.1)	
High	833 (25.2)	22 (4.7)	234 (38.9)	304 (33.4)	273 (23.0)	
Sedentary behaviour (#42)						0.093
<8 h/day	2753 (73.0)	58 (3.4)	671 (32.1)	902 (31.6)	1122 (32.9)	
≥8 h/day	1121 (27.0)	27 (4.1)	230 (26.6)	359 (32.3)	505 (37.1)	
Smoking status (#0)						<0.001
Never	2946 (71.3)	58 (3.7)	642 (29.6)	942 (30.7)	1304 (36.0)	
Past	318 (7.4)	4 (0.8)	70 (24.7)	103 (33.8)	141 (40.6)	
Current	652 (21.3)	25 (4.5)	200 (36.8)	234 (34.5)	193 (24.3)	
Passive smoking (#8)						0.025
No	1740 (39.4)	27 (2.6)	388 (29.5)	571 (30.3)	754 (37.6)	
Yes	2168 (60.6)	59 (4.2)	520 (31.4)	708 (32.8)	881 (31.5)	
Ever used alcohol (#2)						0.058
No	3820 (97.6)	86 (3.7)	891 (30.8)	1239 (31.4)	1604 (34.0)	
Yes	94 (2.4)	1 (0.5)	20 (25.9)	40 (46.4)	33 (27.2)	
Hypertension (#27)						<0.001
No	2183 (63.8)	76 (2.9)	661 (30.3)	733 (31.3)	713 (35.5)	
Yes	1706 (36.2)	11 (0.2)	246 (11.8)	536 (33.0)	913 (55.0)	
Type 2 diabetes (#204)						<0.001
No	3116 (86.0)	74 (1.6)	761 (21.9)	1022 (32.9)	1259 (43.6)	
Yes	596 (14.0)	6 (1.7)	108 (17.4)	188 (28.8)	294 (52.1)	
Heart attack or stroke (#5)						<0.001
No	3670 (95.5)	84 (3.7)	875 (31.6)	1191 (31.4)	1520 (33.3)	
Yes	241 (4.5)	2 (0.7)	37 (13.3)	86 (39.3)	116 (46.8)	

**Table 2 ijerph-18-04198-t002:** Body weight classification by age group and sex among adults in Iraq, 2015.

Variable	General Weight Status (Body Mass Index)				Central Obesity	*p*-Value
	Underweight	Normal Weight	Overweight	Obesity		
	%	%	%	%	%	
Age in years, female						<0.001
18−29	4.6	40.2	31.2	24	44.3	
30−44	0.6	16.5	32	50.9	80.6	
45−59	0.7	8.4	26.6	64.2	91.8	
60−69	0.8	19.2	29.2	50.8	86.9	
Age in years, male						<0.001
18−29	8.5	51.5	25	15	15.5	
30−44	1.6	23.2	42.5	32.7	38.4	
45−59	0.7	17.4	40.5	41.4	56.4	
60−69	1.4	25.3	37	36.3	56.1	

**Table 3 ijerph-18-04198-t003:** Multivariable multinomial logistic regression with overweight and obesity (with under or normal weight as reference category).

Variable	Overweight		Obesity	
	Adjusted RRR (95% CI) ^a^	*p*-Value	Adjusted RRR (95% CI) ^a^	*p*-Value
Age in years				
18−39	1 (Reference)		1 (Reference)	
40−49	2.66 (1.98, 3.57)	<0.001	4.47 (3.39, 5.91)	<0.001
≥50	1.20 (0.81, 1.80)	0.362	1.41 (0.90, 2.21)	0.131
Sex				
Female	1 (Reference)		1 (Reference)	
Male	0.73 (0.53, 1.00)	0.051	0.47 (0.33, 0.68)	<0.001
Education				
<Primary	1 (Reference)		1 (Reference)	
Primary	1.04 (0.75, 1.45)	0.79	0.87 (0.63, 1.19)	0.372
>Primary	0.96 (0.71, 1.29)	0.77	0.69 (0.50, 0.94)	0.02
Adult household members				
1−2	1 (Reference)		1 (Reference)	
3−4	0.62 (0.45, 0.83)	0.002	0.48 (0.35, 0.65)	<0.001
≥5	0.57 (0.43, 0.74)	<0.001	0.45 (0.33, 0.60)	<0.001
Residence				
Rural	1 (Reference)		1 (Reference)	
Urban	1.45 (1.08, 1.95)	0.014	1.58 (1.14, 2.18)	0.006
Meals outside home				
0	1 (Reference)		1 (Reference)	
1	1.28 (0.94, 1.76)	0.118	1.30 (0.92, 1.85)	0.14
≥2	1.01 (0.71, 1.44)	0.938	1.29 (0.88, 1.90)	0.194
Physical activity				
Low	1 (Reference)		1 (Reference)	
Moderate	1.10 (0.81, 1.49)	0.533	1.03 (0.76, 1.40)	0.852
High	0.96 (0.69, 1.32)	0.784	0.72 (0.50, 1.04)	0.082
Smoking status				
Never	1 (Reference)		1 (Reference)	
Past	1.13 (0.62, 2.05)	0.691	1.23 (0.75, 2.02)	0.406
Current	1.00 (0.71, 1.39)	0.991	0.72 (0.48, 1.08)	0.11
Passive smoking				
No	1 (Reference)		1 (Reference)	
Yes	1.07 (0.83, 1.39)	0.595	0.99 (0.73, 1.33)	0.944
Hypertensive				
No	1 (Reference)		1 (Reference)	
Yes	2.18 (1.62, 2.94)	<0.001	3.13 (2.36, 4.17)	<0.001
Type 2 diabetes				
No	1 (Reference)		1 (Reference)	
Yes	1.07 (0.72, 1.60)	0.73	1.18 (0.77, 1.81)	0.446
Heart attack or stroke				
No	1 (Reference)		1 (Reference)	
Yes	1.49 (0.86, 2.59)	0.154	1.40 (0.83, 2.35)	0.204

RRR = relative risk ratio; CI = confidence interval. ^a^ Adjusted for all variables in this table.

## Data Availability

The data for the current study are publicly available from the World Health Organization NCD Microdata Repository (URL: https://extranet.who.int/ncdsmicrodata/index.php/catalog).

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
