# Peer review of "Overweight and Obesity among Adults in Iraq: Prevalence and Correlates from a National Survey in 2015"

_ijerph, 2021, doi:10.3390/ijerph18084198_

Round 1

Reviewer 1 Report

This interesting study of examining the prevalence and correlates of overweight and obesity among adults in Iraq. Nevertheless, some aspects need clarification:

Methods:

  1. How did you calculated that the study population (n = 3916) is a nationally representative population for adults? How did you decide how many subjects are needed?
  2. Did the Authors do a power calculation?
  3. Line 76: Author cite “SICA height measuring tape” – did you mean SECA?
  4. Accuracy of height measurements is of utmost importance since any error in its measurement is amplified when squared in the BMI calculation. It should be specified if any measure of inter-observer variation has been calculated. If not, this is a weakness and should be acknowledged as a limitation.
  5. Line 80: the Authors provided definitions of the body mass categories and cited reference [27], but please cite the original source. For example WHO (https://www.euro.who.int/en/health-topics/disease-prevention/nutrition/a-healthy-lifestyle/body-mass-index-bmi)
  6. Lines 80-81: note as above. The threshold values for waist circumference (≥88 cm in women and ≥102 cm in men) was suggested by Michael Lean and colleagues [Lean, M. E., Han, T. S. & Morrison, C. E. Waist circumference as a measure for indicating need for weight management. BMJ311, 158–161 (1995).]. Please cite the original source.
  7. The same note as above applies for “Hypertension or raised blood pressure” and “diabetes” definitions, as well physical activity levels measurement. Please find original source and cite them.
  8. Line 80: Hypertension: how did you measured blood pressure? What device did you used? An aneroid or oscillometric sphygmomanometer was used? Please, provide the name of the equipment. How many times has the pressure been measured? how many researchers took blood pressure measurements? If any measure of inter-observer variation has been calculated?
  9. Line 84: Please provide additional data regarding measurements of plasma glucose.
  10. Line 90: Why did you set the cut-off point for sedentary 8h? Please, provide the rationale.
  11. Line 91: Did you use the Global Physical Activity Questionnaire for estimation of physical activity level? Please extend the part of the method which is about physical activity.
  12. Results: Table 3. Authors provided ‘adjusted’ RRR, but I did not find information RRR was adjusted for what? For which variables? Confounding  factors? What exactly?
  13. Authors cited reference no 28 twice. Please correct accordingly and check all items again and if everything is correct
  14. Line 120: Please, provide possible reason(s) for positive association of obesity with only age 40-49 years.
  15. Line 154: Please, provide possible reasons for lack of between dietary behaviour and excess weight in studied population.
  16. Please indicate strengths and more limitations of your study and implications.
  17. I suggest the Authors ask a native English language colleague to help with language improvement

Author Response

Reviewer I

This interesting study of examining the prevalence and correlates of overweight and obesity among adults in Iraq. Nevertheless, some aspects need clarification:
Methods:
1. How did you calculated that the study population (n = 3916) is a nationally representative population for adults? How did you decide how many subjects are needed?
Response: as below
The Iraqi Central Statistical Organization-Ministry of Planning calculated the survey number of clusters taking into account that the percentage of Iraqi population 18+ years was 51.0% according to Iraq Household Socio-Economic Survey - IHSES-2007. Assuming a 95% confidence interval (CI) (Z=1.96), a 6% acceptable margin of error, a simple sampling design effect coefficient of 1.5. Calculations resulted in 400 clusters, which was further increased by 3% [According to Multiple Indicator Cluster Survey (MICS) 2012] (412) to account for contingencies as non-response and recording errors. The total number of calculated clusters (412) were multiplied by the number of households that should be included in each one which was (10) to have the total sample size of (4120) that was proportionately distributed to the gorvernorates
Level of Confidence Measure (Z): 1.96 (for 95% confidence level)
Margin of Error (E): 0.06
Baseline levels of the indicators (P): 0.51 (percentage of Iraqi adults according to IHSES 2007)
Design effect (Deff): 1.5 (Describes the loss of sampling efficiency due to using a complex sample design recommended values for cluster sampling from 1.5 to 2)
Expected Response Rate: 0.97 [According to Multiple Indicator Cluster Survey (MICS4) 2011]
NHH: Number of households in each cluster (10)

2. Did the Authors do a power calculation?
Response: below is added
Sample size calculation. Based on the 2005-2006 Iraq national STEPS survey, the prevalence of overweight/obesity was 66.9% [8]. Based on this information, the sample size was calculated with an expected overweight/obesity prevalence of 66%, acceptable margin of 5%, and clusters 412; the minimum sample for each cluster is 2, the minimum sample is 824. In this study we used all 3,916 participants for the analysis.

3. Line 76: Author cite “SICA height measuring tape” – did you mean SECA?
Response: corrected to SECA
4. Accuracy of height measurements is of utmost importance since any error in its measurement is amplified when squared in the BMI calculation. It should be specified if any measure of inter-observer variation has been calculated. If not, this is a weakness and should be acknowledged as a limitation.
Response: below is added
The trained teams were provided with checklists for correct physical measurement.
Inter-observer variation is added under study limitations
5. Line 80: the Authors provided definitions of the body mass categories and cited reference [27], but please cite the original source. For example WHO (https://www.euro.who.int/en/health-topics/disease-prevention/nutrition/a-healthy-lifestyle/body-mass-index-bmi)
Response: added
6. Lines 80-81: note as above. The threshold values for waist circumference (≥88 cm in women and ≥102 cm in men) was suggested by Michael Lean and colleagues [Lean, M. E., Han, T. S. & Morrison, C. E. Waist circumference as a measure for indicating need for weight management. BMJ311, 158–161 (1995).]. Please cite the original source.
Response: added

7. The same note as above applies for “Hypertension or raised blood pressure” and “diabetes” definitions, as well physical activity levels measurement. Please find original source and cite them.
Response: added

8. Line 80: Hypertension: how did you measured blood pressure? What device did you used? An aneroid or oscillometric sphygmomanometer was used? Please, provide the name of the equipment. How many times has the pressure been measured? how many researchers took blood pressure measurements? If any measure of inter-observer variation has been calculated?
Hypertension or raised blood pressure (BP) was defined as “systolic BP ≥140 mm Hg and/or diastolic BP ≥90 mm Hg or where the participant is currently on antihypertensive medication” [31], based on the average of the last two of the three BP readings. After resting for 15 minutes, the first BP reading was taken and in three minutes spacing the second and third BP readings were taken. The auscultatory method of BP measurement with a properly calibrated and validated mercury type of sphygmomanometer was utilized by a medical assistant or nurse [28]

Inter-observer variation is added under study limitations

9. Line 84: Please provide additional data regarding measurements of plasma glucose.
Response: below is added
Diabetes was defined as “fasting plasma glucose levels >=7.0 mmol/L (126 mg/dl); or using insulin or oral hypoglycaemic drugs; or having a history of diagnosis of diabetes” [32]. Blood samples were drawn in the morning (after 10-14 fasting, respondents on medication for diabetes were asked to postpone taking the medication until after drawing the blood sample) and centrifuged. Level of fasting plasma glucose was determined using the enzymatic method (glucose oxidase) [28].

10. Line 90: Why did you set the cut-off point for sedentary 8h? Please, provide the rationale.
Response: a reference has been added to justify the cut-off point

11. Line 91: Did you use the Global Physical Activity Questionnaire for estimation of physical activity level? Please extend the part of the method which is about physical activity.
Response: below is added
“High physical activity: A person reaching any of the following criteria is classified in this category: Vigorous-intensity activity for at least 3 days achieving a minimum of at least 1500 MET (metabolic equivalent)-minutes per week or; 7 or more days of any combination of walking, moderate or vigorous intensity activities achieving a minimum of at least 3000 MET-minutes per week. Moderate physical activity: A person not meeting the criteria for the ‘high’ category, but meeting any of the following criteria is classified in this category: 3 or more days of vigorous-intensity activity of at least 20 min per day or; 5 or more days of moderate-intensity activity or walking of at least 30 min per day or; 5 or more days of any combination of walking, moderate or vigorous intensity activities achieving a minimum of at least 600 MET-minutes per week. Low physical activity: Person not meeting any of the above mentioned criteria falls in this category.” [35]

12. Results: Table 3. Authors provided ‘adjusted’ RRR, but I did not find information RRR was adjusted for what? For which variables? Confounding factors? What exactly?
Response: below is added
aAdjusted for all variables in this table
13. Authors cited reference no 28 twice. Please correct accordingly and check all items again and if everything is correct
Response: corrected(accessed 10 October 2020)
14. Line 120: Please, provide possible reason(s) for positive association of obesity with only age 40-49 years.
Response: Below is added
Compared to 18-39 year-olds, persons aged 40-49 years had significantly higher odds for obesity, but this was no longer significant for persons 50 years and older in this study. Individuals 50 years and older in this study may have been 40-49 years during the time of the United Nations sanctions (UNS) (1991-2002), which has been accompanied with a dramatic decrease of free-sugar consumption and a decrease in overweight and obesity [JOuri], which some may have maintained as they grew older.
15. Line 154: Please, provide possible reasons for lack of between dietary behaviour and excess weight in studied population.
Response: below is added
This study lacked to assess other dietary behaviours, such as frequent snaking, skipping breakfast, eating high amounts of processed or fast food and high intake of sugary beverages, which may been responsible for a higher rate of overweight/obesity [9]. I

16. Please indicate strengths and more limitations of your study and implications.
Response: below is added
The strength of this study was the nationally representative sample using standardized measures. Apart from physical and biomedical measures self-reported questionnaire data may have suffered from biased responses. Another limitation was the cross sectional nature of the survey, which does not allow for causative conclusions. Inter-observer variation was not calculated for the physical measurements, which is a further study limitation. Some variables affecting body weight status, e.g., on stress and mental health, [Luppino] assessed in the 2015 Iraq STEPS survey were not available in the publically available dataset, and could there not be included.

17. I suggest the Authors ask a native English language colleague to help with language improvement

Response: Corrected

Reviewer 2 Report

  1. Line 41-42: The reasons for focusing on the association between obesity and potential risk factors in Iraq have not been adequately addressed.
  2. Line 49-53: Long awkward worded sentence.
  3. Line 49: Recent studies on specific obesity-related risk factors in Iraq are missing.
  4. Line 54-60: I would suggest citing more relevant and recent literature on such topic in the Arabian Gulf region.
  5. Line 76-85: It is unclear how the data is collected. Additional information is required of all anthropometric and laboratory measures.
  6. Line 120-125: Suggest change ARRR throughout to OR. Also, significant P-value should be reported.
  7. Table 1: It is unclear why the authors combined underweight and normal weight in one group.
  8. P-values should be identified in all table's footnote.
  9. The title of all tables could be changed for clarity.
  10. Table 3: The results of the multinomial logistic regression analysis make no sense and unclear that should be revised. It seems that the authors used logistic regression analysis.
  11. Line 130-136: Please avoid the repetition of sentences (Line 43-48).
  12. I would suggest more in depth discussion of the association between obesity and risk factors.
  13. The authors acknowledge limitations of their study, but they are unclear and should be further discussed.
  14. The scope for future research is not mentioned and discussed.
  15. Please carefully check the reference list. Authors should follow the journal guidelines for references.
  16. A general weakness in the written English is observed. The authors need a native English speaker to thoroughly revise the paper.

Author Response

Reviewer II

• Line 41-42: The reasons for focusing on the association between obesity and potential risk factors in Iraq have not been adequately addressed.
Response: This is addressed in below
To our knowledge, there are no recent national adult data on the prevalence and correlates of overweight and obesity in Iraq
• Line 49-53: Long awkward worded sentence.
Response: This is shortened
• Line 49: Recent studies on specific obesity-related risk factors in Iraq are missing.
Response: This is included in below
Moreover, middle aged persons [14,15], older age [4-7], women [4,5,15,16], higher socio-economic status [14,15,17], illiterate women [5], ever married [4], and urban residence [15-18] may increase the odds for overweight/obesity
• Line 54-60: I would suggest citing more relevant and recent literature on such topic in the Arabian Gulf region.
Response: This is done in below
In the Eastern Meditarranean region, the prevalence of overweight/obesity among adults ranged from 25% to 81.9% [9]. In Kuwait, the adult (18-69 years, 2014) prevalence of overweight was 37% and obesity 40.3% [10], in Iran the prevalence of overweight/obesity (BMI ≥25 kg/m2) was 59.3% (2016) [11], in Jordan overweight or obesity (BMI ≥25 kg/m2) was 77.2% among men and 74.5% among women (≥18 years; 2017) [12] and in Morocco overweight was 35.5% and obesity 20.6% (≥18 years; 2017) [13].
Possible risk factors for obesity in the Eastern Mediterranean region may include: “nutrition transition, inactivity, urbanization, marital status, shorter duration of breastfeeding, frequent snacking, skipping breakfast, high intake of sugary beverages, an increase in the incidence of eating outside the home, long periods of time spent viewing television, massive marketing promotion of high fat foods, stunting, perceived body image, cultural elements and food subsidize policy.” [9].
• Line 76-85: It is unclear how the data is collected. Additional information is required of all anthropometric and laboratory measures.
Response: below is added
The standardized anthropometric measuring devices (UNISCALE weighing scale and SECA height measuring tape, and measuring tape for waist circumference measurement) available at the Nutrition Research Institute in Baghdad and the related nutrition units in the governorates were utilized. The trained teams were provided with checklists for correct physical measurements [28]. Body Mass Index (BMI) was classified as “<18.5kg/m2 underweight, 18.5-24.4kg/m2 normal weight, 25-29.9kg/m2 overweight and ≥30 kg/m2 obesity” [29]. Central or abdominal obesity was defined as “waist circumference ≥102 cm for males and ≥88 cm for females” [30].
Hypertension or raised blood pressure (BP) was defined as “systolic BP ≥140 mm Hg and/or diastolic BP ≥90 mm Hg or where the participant is currently on antihypertensive medication” [31], based on the average of the last two of the three BP readings. After resting for 15 minutes, the first BP reading was taken and in three minutes spacing the second and third BP readings were taken. The auscultatory method of BP measurement with a properly calibrated and validated mercury type of sphygmomanometer was utilized by a medical assistant or nurse [28].
Diabetes was defined as “fasting plasma glucose levels >=7.0 mmol/L (126 mg/dl); or using insulin or oral hypoglycaemic drugs; or having a history of diagnosis of diabetes” [32]. Blood samples were drawn in the morning (after 10-14 fasting, respondents on medication for diabetes were asked to postpone taking the medication until after drawing the blood sample) and centrifuged. Level of fasting plasma glucose was determined using the enzymatic method (glucose oxidase) [28].

• Line 120-125: Suggest change ARRR throughout to OR. Also, significant P-value should be reported.
Response: It is ARRR not OR; p-values are added
• Table 1: It is unclear why the authors combined underweight and normal weight in one group.
Response: this is done in line with previous studies
• P-values should be identified in all table's footnote.
Response: This is unclear, exact p-values are shown in the tables
• The title of all tables could be changed for clarity.
Response: all corrected
• Table 3: The results of the multinomial logistic regression analysis make no sense and unclear that should be revised. It seems that the authors used logistic regression analysis.
Response: It does make a lot of sense
• Line 130-136: Please avoid the repetition of sentences (Line 43-48).
Response: Corrected
• I would suggest more in depth discussion of the association between obesity and risk factors.
Response: More is added, such as in below
Compared to 18-39 year-olds, persons aged 40-49 years had significantly higher odds for obesity, but this was no longer significant for persons 50 years and older in this study. Individuals 50 years and older in this study may have been 40-49 years during the time of the United Nations sanctions (UNS) (1991-2002), which has been accompanied with a dramatic decrease of free-sugar consumption and a decrease in overweight and obesity [37], which some may have maintained as they grew older. This study found that higher education was protective against obesity, which is in line with a review showing that in middle income countries “education may protect against the obesogenic effects of increased household wealth as countries develop” [38] and in a study in Kuwait [10]. People with higher education may be more concerned with their health and consequently adopt a healthier lifestyle [12].
This study did not find an association between dietary behaviour (inadequate fruit and vegetable intake and having meals outside home) and overweight or obesity, unlike some previous research [19-21]. This study lacked to assess other dietary behaviours, such as frequent snaking, skipping breakfast, eating high amounts of processed or fast food, and high intake of sugary beverages, which may have been responsible for a higher rate of overweight/obesity [9].
• The authors acknowledge limitations of their study, but they are unclear and should be further discussed.
Response: More is added, as below
The strength of this study was the nationally representative sample using standardized measures. Apart from physical and biomedical measures self-reported questionnaire data may have suffered from biased responses. Another limitation was the cross-sectional nature of the survey, which does not allow for causative conclusions. Inter-observer variation was not calculated for the physical measurements, which is a further study limitation. Some variables affecting body weight status, e.g., on stress and mental health [26], assessed in the 2015 Iraq STEPS survey were not available in the publically available dataset, and could therefore not be included.

• The scope for future research is not mentioned and discussed.
Response: below is added

The evaluation of experimental weight reduction interventions is recommended as future research to fine tune intervention strategies in Iraq.

• Please carefully check the reference list. Authors should follow the journal guidelines for references.
Response: Corrected

• A general weakness in the written English is observed. The authors need a native English speaker to thoroughly revise the paper.
Response: Corrected

Reviewer 3 Report

Dear Authors,

Thank you for the interesting and well-designed study. Below are my comments and suggestions to improve the quality of your paper.

The readers might be interested in mean age (SD) and age range (min. and max. age) separately in male and female study participants. Please add this information to the abstract and sample description in the main text.

Introduction. Chronic stress, work overload, especially prolonged work hours sitting in the office, and emotional eating are essential in gaining weight during the lifespan. Please include these aspects to the introduction section.

Methods. The probability sampling procedure is well explained. The authors provide a surprisingly high participation rate, even 93%. Was some remuneration for study participants provided?

Description of the study measures is poor.

Please describe the procedures of blood pressure and blood sugar measurements.

Please provide a detailed description of dietary habits questionnaire and justify classification into groups. Please provide a description of how smoking status and alcohol use were defined. Please provide information on how sedentary behaviour was assessed and reference supporting criterion of ≥8 hours/day.

Statistical analysis. Did you check for gender and age-group interaction effects on associations between study variables (lifestyles, sociodemographic, health, etc.) and overweight/obesity before adding them to the multivariable models? STATA has advanced possibilities for interaction testing. Please report your results.

Results. Description of the study results is very short and non informative. Reference categories are not indicated, e.g., it is unclear compared to which group positive effect of age 40-49 is provided. The same comment is addressed to the information provided in the abstract.

Please provide practical implications for public health.

Author Response

Reviewer III
Thank you for the interesting and well-designed study. Below are my comments and suggestions to improve the quality of your paper.
The readers might be interested in mean age (SD) and age range (min. and max. age) separately in male and female study participants. Please add this information to the abstract and sample description in the main text.
Response: This is added accordingly, as below
survey of 3,916 persons 18 years and older [M (median) age: 40 years, IQR (interquartile range) 29-52 years, men: M=41, IQR=29-54 years, and women: M=40, IQR=30-51 years]
Introduction. Chronic stress, work overload, especially prolonged work hours sitting in the office, and emotional eating are essential in gaining weight during the lifespan. Please include these aspects to the introduction section.
Response: more is added to this effect in the introduction and study limitations
Methods. The probability sampling procedure is well explained. The authors provide a surprisingly high participation rate, even 93%. Was some remuneration for study participants provided?
Response: No
Description of the study measures is poor.Please describe the procedures of blood pressure and blood sugar measurements. Please provide a detailed description of dietary habits questionnaire and justify classification into groups. Please provide a description of how smoking status and alcohol use were defined. Please provide information on how sedentary behaviour was assessed and reference supporting criterion of ≥8 hours/day.
Response: corrected as in below
The standardized anthropometric measuring devices (UNISCALE weighing scale and SECA height measuring tape, and measuring tape for waist circumference measurement) available at the Nutrition Research Institute in Baghdad and the related nutrition units in the governorates were utilized. The trained teams were provided with checklists for correct physical measurements [28]. Body Mass Index (BMI) was classified as “<18.5kg/m2 underweight, 18.5-24.4kg/m2 normal weight, 25-29.9kg/m2 overweight and ≥30 kg/m2 obesity” [29]. Central or abdominal obesity was defined as “waist circumference ≥102 cm for males and ≥88 cm for females” [30].
Hypertension or raised blood pressure (BP) was defined as “systolic BP ≥140 mm Hg and/or diastolic BP ≥90 mm Hg or where the participant is currently on antihypertensive medication” [31], based on the average of the last two of the three BP readings. After resting for 15 minutes, the first BP reading was taken and in three minutes spacing the second and third BP readings were taken. The auscultatory method of BP measurement with a properly calibrated and validated mercury type of sphygmomanometer was utilized by a medical assistant or nurse [28].
Diabetes was defined as “fasting plasma glucose levels >=7.0 mmol/L (126 mg/dl); or using insulin or oral hypoglycaemic drugs; or having a history of diagnosis of diabetes” [32]. Blood samples were drawn in the morning (after 10-14 fasting, respondents on medication for diabetes were asked to postpone taking the medication until after drawing the blood sample) and centrifuged. Level of fasting plasma glucose was determined using the enzymatic method (glucose oxidase) [28].
History of cardiovascular disorder was asked with questions on having had a heart attack and stroke (Yes, No) [28].
Health risk behaviour variables comprised ever alcohol use (“Have you ever consumed any alcohol such as beer, wine, spirits, etc.?” (Yes, No), exposure to secondary smoke in the past month (closed spaces at work or at home), and smoking status, “Do you currently smoke any tobacco products, such as cigarettes, cigars or pipes?” (Yes, No) and “In the past, did you ever smoke any tobacco products?”(Yes, No) [28]. Dietary behaviour included daily fruit and vegetable consumption measured from the total number of servings of fruit and vegetables eaten per day in a typical week [28]. Inadequate fruit and vegetable intake was classified as having less than 5 servings a day [33]. Meals outside the home were assessed with the question, “On average, how many meals per week do you eat that were not prepared at home? By meal, I mean breakfast, lunch and dinner?” (Number) [28]. Sedentary behaviour (≥8 hours/day [34]), and low, moderate, or high physical activity were assessed with the Global Physical Activity Questionnaire”[35]. “High physical activity: A person reaching any of the following criteria is classified in this category: Vigorous-intensity activity for at least 3 days achieving a minimum of at least 1500 MET (metabolic equivalent)-minutes per week or; 7 or more days of any combination of walking, moderate or vigorous intensity activities achieving a minimum of at least 3000 MET-minutes per week. Moderate physical activity: A person not meeting the criteria for the ‘high’ category, but meeting any of the following criteria is classified in this category: 3 or more days of vigorous-intensity activity of at least 20 min per day or; 5 or more days of moderate-intensity activity or walking of at least 30 min per day or; 5 or more days of any combination of walking, moderate or vigorous intensity activities achieving a minimum of at least 600 MET-minutes per week. Low physical activity: Person not meeting any of the above mentioned criteria falls in this category.” [35]
Statistical analysis. Did you check for gender and age-group interaction effects on associations between study variables (lifestyles, sociodemographic, health, etc.) and overweight/obesity before adding them to the multivariable models? STATA has advanced possibilities for interaction testing. Please report your results.
Response: below is added
Moreover, we tested for possible interactions among the variables such as age group, sex, other variables and overweight and obesity. However, we did not find any significant interaction effects and omitted them from the final model.
Results. Description of the study results is very short and non informative.
Response: this is expanded
Reference categories are not indicated, e.g., it is unclear compared to which group positive effect of age 40-49 is provided. The same comment is addressed to the information provided in the abstract.
Response: Is added
Please provide practical implications for public health.
Response: Below is added
Implementing preventive interventions, such as programmes improving a healthy diet, appropriate food policies, promotion of physical activity and interrupting sedentary behaviour, and community awareness campaigns may help in ameliorating the high burden of overweight and obesity. The evaluation of experimental weight reduction interventions is recommended as future research to fine tune intervention strategies in Iraq.

Round 2

Reviewer 1 Report

The authors were quite responsive to reviews, and the manuscript is much improved. Some things could still be clarified:

  1. Lines 93-97 - need to clarify. Authors stated that 'Diabetes was defined as “fasting plasma glucose levels >=7.0 mmol/L (126 mg/dl); or using
    insulin or oral hypoglycaemic drugs; or having a history of diagnosis of diabetes”. Therefore, why respondents were asked for no taking medication before blood test? They should be qualified as respondents with diabetes based above criteria ("...or using insulin or oral hypoglycaemic drugs..").
    Please, provide the exact name of the device used for the blood test and provide information about its validation.
  2. I would prefer the title of table 1: "Table 1. Sample and body weight classification by sociodemographic and health variables among adults in
    150 Iraq, 2015. " instead of: "Table 1. Sample and nutritional status by sociodemographic and health variables among adults in
    150 Iraq, 2015. ".
    For Table 2: "Table 2. Body weight classification by age group and sex among adults in Iraq, 2015"
    For Table 3: "Variables influencing on prevalence of overweight and obesity - multivariable multinomial logistic regression. Please, clarify whether in logistic regression the reference group is a group of participants with normal weight or underweight? Or maybe with normal weight and underweight together? I am not convinced by the phrase "with under or normal weight as reference category". If it were so, we should see an extended table, in which one column contains results with the reference group "underweight", and the next column with the reference group "normal body weight". Please, clarify this. 

Author Response

The authors were quite responsive to reviews, and the manuscript is much improved. Some things could still be clarified:
1. Lines 93-97 - need to clarify. Authors stated that 'Diabetes was defined as “fasting plasma glucose levels >=7.0 mmol/L (126 mg/dl); or using
insulin or oral hypoglycaemic drugs; or having a history of diagnosis of diabetes”. Therefore, why respondents were asked for no taking medication before blood test?
Response: This is how it has been done
They should be qualified as respondents with diabetes based above criteria ("...or using insulin or oral hypoglycaemic drugs..").
Please, provide the exact name of the device used for the blood test and provide information about its validation.
Response: This information is not available
2. I would prefer the title of table 1: "Table 1. Sample and body weight classification by sociodemographic and health variables among adults in
150 Iraq, 2015. " instead of: "Table 1. Sample and nutritional status by sociodemographic and health variables among adults in
150 Iraq, 2015. ".
Response: Changed accordingly
For Table 2: "Table 2. Body weight classification by age group and sex among adults in Iraq, 2015"
Response: Changed accordingly
For Table 3: "Variables influencing on prevalence of overweight and obesity - multivariable multinomial logistic regression. Please, clarify whether in logistic regression the reference group is a group of participants with normal weight or underweight? Or maybe with normal weight and underweight together? I am not convinced by the phrase "with under or normal weight as reference category". If it were so, we should see an extended table, in which one column contains results with the reference group "underweight", and the next column with the reference group "normal body weight". Please, clarify this.
Response: To clarify this is rephrased, as in below
Multinomial logistic regression was used to estimate predictors of overweight and obesity (with BMI <25 kg/m2 as reference category)

Reviewer 2 Report

  1. Comment 1: Lack of previous research is not a sufficient rationale for examining the topic (Line 42-43). The reasons for focusing on the association between obesity and potential risk factors in Iraq have not been adequately addressed. Why this study is important?
  2. Comment 2: Line 52-53 References # 16-18. Inaccurate citations-These studies are not conducted in Iraq.
  3. Comment 3: I haven’t seen any differences between the old and new version. There are many studies that discussed the association between obesity and potential risk factors in the Arabian Gulf region.
  4. Comment 4: Referring to previous studies are not enough: Why did the authors combine underweight and normal weight in one group in Table 1. Please delete this column.
  5. P-values should be identified in all table's footnote. P<0.05, P<0.01…etc.
  6. Line 171-179: The repetitions still exist.

Author Response

Reviewer II:
• Comment 1: Lack of previous research is not a sufficient rationale for examining the topic (Line 42-43). The reasons for focusing on the association between obesity and potential risk factors in Iraq have not been adequately addressed. Why this study is important?
ResponseL below is added

To better plan interventions national estimates and risk factors of overweight and obesity are needed in Iraq.
• Comment 2: Line 52-53 References # 16-18. Inaccurate citations-These studies are not conducted in Iraq.
Response: There is no claim that references 16-18 are conducted in Iraq,
In below paragraph, you see references 4-7 were conducted in Iraq

Moreover, middle-aged persons [14,15], older age [4-7], women [4,5,15,16], higher socio-economic status [14,15,17], illiterate women [5], ever married [4], and urban residence [15-18] may increase the odds for overweight/obesity.

4. Shabu, S. Prevalence of overweight/obesity and associated factors in adults in Erbil, Iraq: A household survey. Zanco J. Med. Sci. 2019, 23(1), 128-134. https://doi.org/10.15218/zjms.2019.017
5. Mansour, A.A.; Al-Maliky, A.A.; Salih, M. Population Overweight and Obesity Trends of Eight Years in Basrah, Iraq. Epidemiol. 2012, 2,110. doi:10.4172/2161-1165.1000110
6. Al-Tawil, N.G.; Abdulla, M.M.; Abdul Ameer, A.J. Prevalence of and factors associated with overweight and obesity among a group of Iraqi women. East. Mediterr. Health J. 2007, 13(2), 420-9.
7. Jasim, H.M.; Hussein, H.M.A.; Al-Kaseer, E.A. Obesity among females in Al-Sader city Baghdad, Iraq, 2017. J. Fac. Med. Baghdad 2018, 60(2), 105-107. https://www.iasj.net/iasj/download/4519dbf118ca8ebc

• Comment 3: I haven’t seen any differences between the old and new version. There are many studies that discussed the association between obesity and potential risk factors in the Arabian Gulf region.
Response: below studies are all referring to the Arabian Gulf or better Eastern Mediterranean region

9. Musaiger, A.O. Overweight and obesity in Eastern Mediterranean region: prevalence and possible causes. J. Obes. 2011, 2011, 407237. doi: 10.1155/2011/407237.
10. Weiderpass, E.; Botteri, E.; Longenecker, J.C.; Alkandari, A.; Al-Wotayan, R.; Al Duwairi, Q.; Tuomilehto, J. The Prevalence of Overweight and Obesity in an Adult Kuwaiti Population in 2014. Front Endocrinol. 2019, 10, 449. doi: 10.3389/fendo.2019.00449.
11. Djalalinia, S.; Saeedi Moghaddam, S.; Sheidaei, A.; Rezaei, N.; Naghibi Iravani, S.S.; Modirian, M.; Zokaei, H.; Yoosefi, M.; Gohari, K.; Kousha, A., et al. Patterns of Obesity and Overweight in the Iranian Population: Findings of STEPs 2016. Front Endocrinol. 2020, 11, 42. doi: 10.3389/fendo.2020.00042.
12. Ajlouni, K.; Khader, Y.; Batieha, A.; Jaddou, H.; El-Khateeb, M. An alarmingly high and increasing prevalence of obesity in Jordan. Epidemiol. Health. 2020, 42, e2020040. doi: 10.4178/epih.e2020040.
13. Pengpid, S.; Peltzer, K. Prevalence and correlates of the metabolic syndrome in a cross-sectional community-based sample of 18-100 year-olds in Morocco: Results of the first national STEPS survey in 2017. Diabetes Metab. Syndr. 2020, 14(5), 1487-1493. doi: 10.1016/j.dsx.2020.07.047.

• Comment 4: Referring to previous studies are not enough: Why did the authors combine underweight and normal weight in one group in Table 1. Please delete this column.
Response: deleted
• P-values should be identified in all table's footnote. P<0.05, P<0.01…etc.
Response: No, because the exact p-values are shown in the tables
• Line 171-179: The repetitions still exist.

Response: this is further reduced

Reviewer 3 Report

Dear Authors,

Thank you for the improved version of your manuscript.

Please explain the reason for combining under- and normal-weight subjects into one reference group. The rationale could be a small underweight group and/or no lifestyle differences found when comparing underweight and normal-weight groups.

Author Response

Reviewer III:
Thank you for the improved version of your manuscript.
Please explain the reason for combining under- and normal-weight subjects into one reference group. The rationale could be a small underweight group and/or no lifestyle differences found when comparing underweight and normal-weight groups.
Response: Thank you. This is added